# Mindfulness-Based Stress Reduction (MBSR) and Self Compassion (SC) Training for Parents of Children with Autism Spectrum Disorders: A Pilot Trial in Community Services in Spain

**DOI:** 10.3390/children8050316

**Published:** 2021-04-21

**Authors:** Liliana Paulina Rojas-Torres, Yurena Alonso-Esteban, María Fernanda López-Ramón, Francisco Alcantud-Marín

**Affiliations:** Department of Developmental and Educational Psychology, University of Valencia, 46010 Valencia, Spain; liliana_ro_6@hotmail.com (L.P.R.-T.); Yurena.Alonso@uv.es (Y.A.-E.); M.Fernanda.Lopez@uv.es (M.F.L.-R.)

**Keywords:** mindfulness, autism spectrum disorders, parental stress, parent anxiety

## Abstract

This study aims to develop a clinical trial to test the efficacy of a mindfulness-based stress reduction (MBSR) and self-compassion (SC) program on self-reported values of anxiety, depression, and stress in parents of children with autism spectrum disorder (ASD) in primary school, in order to assess their integration into the framework of community intervention programs in Spain. Methods: A brief 8-week training program using mindfulness-based intervention (MBSR) and self-compassion (SC) has been applied to twelve Valencian ASD parents, ten of whom completed the program. Participants were assigned to two groups; both groups received the same treatment but at two different measurement moments. Depression, anxiety, stress, satisfaction with life and mindful attention awareness measurements were performed, in all participants, in three testing stages. Results: Analysis of variance results suggested that MBSR and SC training reduces stress and anxiety and increases mindful attention awareness. No significant changes were found in life satisfaction measurements. Conclusions: The small number of participants prevents us from generalizing the results found. More MBSR and SC clinical trials are needed in parents of ASD with results on anxiety, depression and stress in order to demonstrate the relevance of the inclusion of these programs in community-based early intervention services.

## 1. Introduction

In the last decade, interest in mindfulness effects studies has increased significantly, as reflected in the number of recent publications [1,2]. Mindfulness is a theoretical construct that can be understood as the capacity related to mindful attention, and the attentive and reflective being in a non-judgmental way [3]. To analyze the positive effects produced by the development in mindful capacity, numerous clinical trials and intervention programs have been developed, as exposed in successful meta-analysis studies [4,5,6,7,8].

Likewise, neurological studies that show the relationship with mindfulness practice are especially relevant. For instance, in meta-analysis research, Falcone and Jerram [9] found that with meditation practice, brain activity increased in the frontal, anterior and insular regions, showing different results in experienced meditation practitioners compared to those who were not experienced in meditation. They also studied the possible genetic mechanisms that underlie the oxytocin receptors and their relationship with executive functions and the empathic network (including the right angular gyrus, the medial prefrontal cortex and the anterior cingulate cortex) and the development of self-compassionate attention [10]. These findings, still preliminary, advance our understanding of how the improvement of mindfulness skills can enhance a person’s well-being and the prevention of any related psychological or physical conditions.

The mindfulness-based intervention techniques that have received the majority of support from the research community are the ones that aim to achieve stress reduction [11]. It is worth noting the two main research lines followed: MBSR (mindfulness-based stress reduction; e.g., [12]) and MBCT (mindfulness-based cognitive therapy; e.g., [13]). Both mindfulness-based approaches can be classified as brief therapies (8 sessions), typically including mindfulness-based practices along with other stress reduction and cognitive therapy approaches. Habitually, these interventions mainly focus on objectives that are centered on teaching participants to observe, recognize and let go of their judgmental thoughts, feelings and emotions that come to mind during mindfulness practice [14,15,16,17]. Although mindfulness techniques have their roots in the Eastern Vipassana meditation tradition (for a review, see [18]), they are also part of the theory and practice of dialectical behavior therapy (DBT; [19]), person-based cognitive therapy (PBCT; [20,21]) and acceptance and commitment therapy (ACT; [22]). Recently, the term “Mindfulness Integrated Cognitive Behavioural Therapy” (MiCBT) was created [23] in order to make explicit the integration of the theoretical principles of both mindfulness and cognitive-based techniques.

Parenting in modern society represents a source of stress for parents who must adjust their work responsibilities to those of upbringing. Consequently, psychological distress and discomfort can be found at the base of multiple alterations in family life and child-rearing [24]. Parents who are facing the demands of raising a child with some type of developmental disorder cope with a greater number of stress sources [25,26] that are present from the first developmental disorder-related warning signs [27]. Parents often report disturbances in their psychological well-being [28,29,30,31] or high levels of stress [32,33]. For instance, one of the most documented consequences of high stress is altered patterns of parent–child interaction [34,35]. It is worth mentioning that sources of stress experienced by parents are not only due to the changes that they must make to the family dynamic or the amount of time that they must invest in meeting their children’s needs, but also due to their participation in intervention or treatment activities [36]. Parental stress also feeds back into personal stress, having consequences not only for parents (i.e., on their personal stability and family relationships, e.g., [37]) but also has a clear impact on children’s psychological well-being and can cause a possible reduction in the effectiveness of an intervention or treatment programs (e.g., [28]).

Therefore, many mindfulness-focused intervention programs aim to develop parents’ or primary caregivers’ psychological well-being [38,39]. By optimizing psychological happiness in parents or caregivers, we can obtain benefits for the well-being of children diagnosed with autism spectrum disorders (ASD) [28]. It must be pointed out that there are two main types of parents of children with ASD intervention-focused therapy: the interventions that need to be done with parents only and the ones that are intended to be applied simultaneously with parents and children (i.e., aimed at solving parent–child interaction problems). In the parents-only group, PCIT (parent–child interaction therapy; [40]), PACT (preschool autism communication trial; [41]) and TP (Triple P: Positive Parenting Program; Sanders, [42,43]) are some examples that can be noted. On the other hand, the second group of parents’ interventions regarding mindfulness programs combines approaches addressed not only towards children and youth with ASD but also to their parents or caregivers. For instance, MYmind [44] is a mindfulness-based program developed specifically for young people diagnosed with neurodevelopmental disorders and their parents. In this program, children and youths with ASD, and their parents, follow parallel sessions in which they practice mindfulness meditation exercises and then develop the ability to apply them in difficult situations [45]. As an example, Ridderinkhoff et al. [46] studied a group of forty 9-to 17-year-old children (and their parents). They applied MYmind, proposing an interdependence in the model between abilities such as knowing and connecting with peers, pausing their impulses, being aware of the present moment, and letting go in a nonjudgmental way, coping strategies, responding calmly to others’ demands, etc. Similarly, Salen-Guirgis et al. [47] studied a 23-day parent–child trial, finding improvements in ASD symptoms, emotional regulation and adaptive skills in young people and their parents. According to the data, MYmind has proven to be a program that can lead to an improvement in emotional regulation and the adaptability of young people with autism and parental care.

With regard to the aforementioned parent intervention methodologies, it can be noted that recent research has also shown that mindfulness-based programs have achieved positive changes in parents of children with neurodevelopmental disorders [28]. In particular, these programs offer benefits for stress and anxiety coping abilities in parents of children with ASD [28,36,37,48]. Overall, these studies showed positive improvements in the parents’ well-being by reducing stress and increasing their levels of happiness [12]. They also report that caregivers exhibited an improvement in positive children’s demand responses in the sense that their feedback was found to be more empathetic and appropriate, with greater levels of focused attention, cognitive flexibility and emotional regulation, all of which are abilities that are trained and reinforced in mindfulness-based interventions [48].

In the same way, Singh et al. [49] used a modified version of MBSR named Mindfulness-Based Positive Behavior Support (MBPBS). They found better results by applying the therapeutic combination of MBSR and MBPBS than by using them separately. These results have been verified not only for families with children with ASD, but also for ID (intellectual disabilities) children’s families. In summary, previous scientific studies that have applied intervention programs based on contemplative practices (e.g., mindfulness-based stress reduction or variants, self-compassion) or based on emotional regulation (e.g., [44]) have focused primarily on developing psychological well-being for parents or primary caregivers of children with ASD by studying the impact of mindfulness-based training on either stress, anxiety or depression, but have not studied all three aspects at once in a sample of ASD children’s parents or caregivers. Therefore, the clearest background results have been reported with stress factors. Cheung, Leung and Mak [50] found that parental stress could be seen as a mediating factor in relation to the exercise of more mindfulness-focused parenting that could affect the internalization of existing social stigmas concerning autism. On the other hand, Singh et al. [51] found that mothers of adolescents with ASD reduced their stress levels as a result of an intervention in MBSR. Finally, Torbet, Proeve and Roberts [52] found that self-compassion (SC) training could introduce a protective reason for the development of parental stress in parents of children diagnosed with ASD. Secondly, in relation to depression, Blackledge and Hayes [53] found an improvement in symptoms following SC and acceptance training. Finally, in relation to self-reported anxiety, we have found no previous studies that present results in relation to the reduction of anxiety values as a result of a mindfulness-based intervention. Therefore, and in order to fill the existing gap in earlier research, in the present study, we propose to evaluate the impact of mindfulness and self-compassion training on anxiety, depression and stress values, in a sample of parents with children diagnosed with ASD. To our knowledge, there are no studies that evaluate these three psychological aspects in the same sample of ASD children´s parents or caregivers, at ages that are close to the communication of their child’s ASD diagnosis.

Autism spectrum disorders are neurodevelopmental disorders characterized by persistent deficits in communication and social interaction and restricted, repetitive patterns of behaviors, activities and interests that cause significant impairment in social, occupational or other areas of functioning. Such symptoms are typically present in the early developmental period [54]. In children, the development of the nervous system is influenced by the interaction with the environment [55]. A child with ASD has deficits in skills for basic communication and social interaction, which can generate inadequate patterns of interaction, due to stress caused by the new situation, among other reasons [35,56]; this negatively affects neuropsychological development, producing a cascade effect [57,58,59]. Inappropriate interactions have two negative effects: they continue to feed back into the child’s developing nervous system, increasingly diverting it from normative development, and they generate a high level of tension and discomfort in the parents, which in turn feeds the inappropriate interactions [60].

In the present research, we aim to evaluate the effectiveness of a mindfulness-based and self-compassion program on self-reported values of anxiety, depression and stress in parents of children with ASD in primary school. We believe that the results obtained in the present study can be relevant for the inclusion of mindfulness-based interventions as a tool to be used in community center programs in order to help parents coping with stress, depression and anxiety.

## 2. Methods

### 2.1. Procedure

The trial was conducted from February to May of 2019. It was designed with a control group (waiting list). Parents were recruited from a call for participation made to all members of ASPAU (Asociación Proyecto Autismo de Valencia). The inclusion/exclusion conditions were: (a) children under 12 years of age (primary school age); (b) with a diagnosis of ASD, without associated intellectual disability; (c) that the participating parents lived with the children and (d) a signed informed consent form was provided (see Figure 1). This project was approved by the University of Valencia’s Committee on Ethics in Research with Humans, which guarantees compliance with the principles of the Helsinki agreement (Code: H1541505018986).

### 2.2. Participants

Twenty-five parents responded to our call; after the first interview, 13 parents did not meet the eligibility criteria or declined to participate. Participation was voluntary, unpaid and adapted to the participants’ schedules. Due to family responsibilities, only one of the two parents participated in the program. The participants belonged to a medium social economic stratum, with medium and higher education level, and lived with the child with ASD and his or her spouse. The rest (twelve participants) were assigned to two groups according to their time availability (mornings or afternoons), not randomly. Only one participant had minor previous experience in body relaxation practices, and the rest of them had never engaged in meditation, relaxation or yoga exercises before the intervention. It is worth noting that none of the participants had participated in a ruled mindfulness and self-compassion-based program before, and they were all naive to all the methods proposed during the training. The first group participated in the program in January–March, leaving the second group on a waiting list. Once the program was finished for the first group, the intervention was performed in the second group. Two parents (one from each group) did not complete the program due to changes in their work schedule, which made it impossible for them to attend the sessions. The descriptive family data of the participants are shown in Table 1. Only one member of the parent couple was allowed to participate. In group A, 5 mothers completed the program, and in group B, 4 mothers and 1 father completed the program.

### 2.3. Characteristics of Children

All the children were diagnosed with autism spectrum disorder according to DSM-5 [54] and had been diagnosed between 3 and 5 years old. In some cases, due to the child’s age, the official diagnosis was TGD-Unspecified (DSM-IV T/R, APA, [61]). All children attended regular education centers with support (Level 1: “Requiring support”; Level 2: “Requiring substantial support” and Level 3: “Requiring very substantial support” [54]).

### 2.4. Measures

In order to provide evidence of the results of our mindfulness and self-compassion-based program, different evaluation tools commonly used in meditation practice have been analyzed. As many earlier studies reported (e.g., [5]), the mindfulness concept has multidimensional and complex meanings, not only related to the theoretical facet but also on the applied dimensions. Mindfulness conceptualization and measurement are always changing and growing [62]. Accordingly, Sauer et al. [63] reviewed eleven different intervention tools for measuring mindfulness, classifying them into two large groups according to the concept that they rate and whether mindfulness is treated as a one-dimensional or a multidimensional construct. At the same time, and depending on the results, it also determines their suitability depending on whether they evaluate experts or novices in the practice of mindfulness or target specific populations or specific intervention goals. In this regard, one of the most widely used and highly rated instruments is the MASS (Mindful Attention Awareness Scale; [64]), a short questionnaire (15 items) that measures the general tendency to pay attention to or be aware of the present experience in daily life. It has a one-dimensional structure that provides a total score that is the result of the sum of the item scores. The original version reaches an internal consistency of 0.82 (Cronbach’s Alpha). The MASS has been selected for the present study mainly because it is indicated for populations without previous experience in mindfulness training [62]. We used the MASS Spanish version adapted by Soler et al. [65] with an internal consistency of 0.89 (Cronbach’s Alpha).

For measuring anxiety, stress and depression indexes, we used DASS-21 (Depression, Anxiety and Stress Scale; [66]). This version is the short form derived from the 42-item DASS [66]. The DASS-21 is a self-report composed of 21 items, which evaluates the three dimensions (anxiety, stress and depression). The score for each scale is obtained by adding up the scores of each item (multiplying it by 2 to equate it with the score of the long version DASS) and the results for each dimension vary between 0 and 42 points [67]. The DASS-21 has been validated in the Spanish population, showing adequate psychometric properties for the adult population [68] and for the clinical population [67]. For the present study, we used the Spanish version developed by Bados, Solanas and Andrés [69], which showed acceptable internal consistency indices (Cronbach’s Alpha, according to scale: depression 0.84, anxiety 0.70 and stress 0.82).

Lastly, to study the general consequences of the program, we used the SWLS (Satisfaction with Life Scale; [70]) for assessing satisfaction, psychological well-being and happiness dimensions. Previous research [71,72] shows that satisfaction with life correlates with mental health measures and predicts future behaviors, such as suicide attempts. The version used in this study is the Spanish version of SWLS of Atienza, Pons, Balaguer and García-Merita [73], with an internal consistency of 0.84 (Cronbach’s Alpha).

### 2.5. Mindfulness and Self-Compasion-Based Intervention Program

The intervention proposed for the present work was inspired by MBSR [16,74,75] and MSC (mindful self-compassion) training programs [76,77,78] and the characteristics of the sessions were adapted to the particular needs of the population that we were addressing (i.e., parents of children with ASD).

We mainly organized the sessions based on an MBSR standardized program (i.e., based on the integration of meditation techniques, body awareness and yoga and the understanding of the neuropsychological functioning of stress). At the end of the 8 weeks of MBSR training (e.g., [75]), the participant is expected to show an increased ability to manage stress and daily life challenges, face disturbing events in a more adaptive way and increase their sensation of remaining fully present at every moment, all with the goal of improving their well-being. It is worth remarking that prior to mindfulness training, participants reported anxiety as a consequence of the social stigmatization that society places on them and the consequent daily self-criticism that they experienced. This outcome, also verified by previous research (e.g., [50]), encouraged us to include a specific combination of psychological and educational sessions designed to cope with ASD social stigmatization for reducing daily stress. Additionally, we also integrated some aspects that we believed to be fundamental for ASD parents, based on MSC training [79] that focuses on the development of self-compassion skills to overcome self-criticism and prevent them from becoming anxiety or depression. Self-compassion [77] involves the ability to cope without mercilessly judging and criticizing ourselves by learning to be kind, self-comforting and understanding when confronted with personal failings or when feeling frustrated. Additional modifications (following recommendations addressed by [80,81]) were made in aspects of the organization of the sessions (i.e., their duration, environmental conditions, etc.) and in their content to adapt the sessions to the population.

In general terms, the program that we performed consisted of 8 sessions of around 90 min (1 per week). In addition to the face-to-face session, participants were asked to perform exercises at home (between 15 and 30 min per day). Each session was divided into two parts: the first consisted of the development and training on mindfulness exercises, being moderated by one of the authors of this article. The second part consisted of developing psychoeducational sessions about the ASD disorder, based on active listening [82,83], and mediated by one of the authors, an expert in the ASD research area.

According to the recommendations of many current publications of the line inspired by MBSR and MSC (e.g., [80]), the weekly sessions were accompanied by tasks to complete at home. We supported the homework with different resources such as videos, readings and targeted practices that aimed to enable participants to build up autonomy and self-management coping tools for everyday situations. Typically, each session had the same pre-set structure order that is detailed as follows: (a) shared group thoughts about videos or documents that were assigned before (duration: 15 min); (b) shared group thoughts about day-to-day homework tasks (duration: 15 min); (c) participant´s individual relevant insights achieved during the accomplishment of the home tasks (duration: 35 min); (d) explanation of the session and homework activities (i.e., formal and informal practices) of the current week (duration: 25 min); (e) joint mindfulness practice conducted by the mindfulness trainer. This last part of the session is designed not only as a mindfulness practice but also as a practical explanation of the formal task of the current week that the participants typically must do on their own before the next joint session (duration: 30 min). The general topics and practices are performed in both the mindfulness and psychoeducational parts of the face-to-face sessions are explained in Table 2.

It is worth mentioning that prior to the mindful and self-compassion training, participants reported anxiety as a consequence of the social stigmatization that society puts on them and their ASD child. This finding is also verified by previous research [50], and it encouraged us to include a specific session to cope with ASD social stigmatization for reducing daily stress.

### 2.6. Weekly/Daily Qualitative Assessment

Following MBSR program recommendations (e.g., [16]), a set of ad hoc daily homework activity sheets were included, specially adjusted to each session. They typically included both formal and informal practice sheets (PS). Formal homework practice consisted of applying at home the methods learned during the previous session, and the informal homework practices involved tasks related to bringing mindful awareness to some daily routine activity. The formal daybook accordingly contained PS for documenting impressions after the formal activity was completed at home. The informal daybook was where the participants were asked to record on the PS any daily activity that could transform into a moment of full attention by applying what was learned in the previous session of each week. Typically, the formal and informal daybook charts had six boxes in order to dedicate one to each day of the week. On the last day of the week, prior to the group session, participants received the multimedia material and preparatory documents for the face-to-face session through the WhatsApp group or by accessing the course website. All participants were required to attend the session having read, heard or seen the material submitted earlier by WhatsApp.

During the first 15 min of each session, the participants expressed their thoughts about the material sent, analyzed the records that they had made, and then the moderator therapist explained the activities for the following week. Once the more explanatory part was concluded, a joint contemplative practice was carried out (i.e., guided meditation, yoga session, etc.) modeled by the therapist and adapted to the aims of each session.

### 2.7. Support Materials

For the development of the program, there was a website (Moodle) that was used as a repository of documents and multimedia material that were used as needed throughout the group sessions and during the individual practice performed by each participant as homework (formal and informal). Fluid and constant communication was also maintained through a WhatsApp group that also allowed the distribution of documents and multimedia material. The material was distributed on a weekly basis, and questions raised by participants were reported or answered on a daily basis.

### 2.8. Adherence to Treatment

We only had two waivers throughout the program (one in each group). One of them (group A) occurred after treatment in the follow-up phase. The second (Group B) occurred during treatment. Both cases were excluded from data analysis. The daily activities that they had to register in the daybooks were used also as a system to monitor adherence to treatment. From the analysis of these records, it could be concluded that adherence and commitment to treatment were satisfactory (i.e., more than 80% of the participants recorded the PS daily and the rest did so in 67% of the sessions). There were no cases of participants that did not accomplish all the weekly sessions.

### 2.9. Design and Data Analysis

The participants were assigned to two crossed groups [84]. In this sense, the first group received the mindfulness and self-compassion-based training, while the second group remained on the waiting list. In the second stage, the first group remained on the waiting list without receiving treatment in the follow-up phase, while the second group received the same mindfulness and self-compassion-based treatment (see Figure 2).

## 3. Results

In Table 3, the means and standard deviations (SD) of the three scales of the DASS-21 (i.e., Stress, Anxiety, and Depression) in the three assessed measurements moments, separated by group, are shown. All the calculations were performed with SPSS, version 26, licensed by the University of Valencia.

Since the assignment to the groups was not random, we performed some specific analyses of variance in order to analyze the homogeneity of the groups of the three measures in relation to the two groups, and we did not observe significant differences, except for DASS-Stress in measures 1 and 3 (F (measure 1) = 11.37, *p* = 0.1; F (measure 3) = 9.53, *p* = 0.02). Additionally, we performed some analyses in relation to the between-subjects variable “GROUP”, obtaining only significant differences in MASS (F = 124.8, *p* = 0.00). In this sense, while no main effects of “GROUP” were found in DASS (i.e., anxiety, depression and SWLS), we assumed that we could perform multivariate analysis to estimate treatment effects. In the case of the DASS-Stress and MAAS questionnaire, because they showed some significant effects on the variable GROUP on the corroborations that we made, we assumed that the observations of the data presented in the present work related to these two variables should be taken as orientating and in future work should be analyzed and supported with a larger sample of data.

As the core analysis of the Results section, we performed a two-way mixed ANOVA, with ASSESSMENT TIME (i.e., measure 1, measure 2, measure 3) as a within-subject factor and GROUP as a between-subject factor. In Table 4, we present the details of the variance model (i.e., two main effects—main effect of ASSESSMENT TIME and main effect of GROUP—and one interaction—ASSESSMENT TIME × GROUP) for each questionnaire.

We additionally performed univariate tests of significance for planned comparisons to see the variances that arose from the combination of the between- and within-subjects factors and their contrast coefficients. In Table 5, the results of the post hoc test (i.e., contrasts) analysis are provided.

The main results of these corroborations are as follows and can be observed graphically in Figure 3. Firstly, concerning the DASS Anxiety scale, regarding the Group A between-subjects factor, DASS Anxiety measurement 1 and DASS Anxiety measurement 2 as within-subjects factors were selected for comparison. In this group, it represented the training phase, and we obtained F(1,8) = 8.74; *MSE* = 18.3; *p* = 0.02. On the other hand, regarding the Anxiety scale, for the Group B between-subject factors, DASS Anxiety measurement 2 and DASS Anxiety measurement 3 were selected as within-subjects factors for comparison. In this group, it represents the training phase, and we obtained F(1,8) = 4.5, *MSE* = 25.8; *p* = 0.06. In summary, regarding the DASS Anxiety scale, we found a significant improvement between the evaluation measurements comprising the training phase (pre–post) in both groups. Secondly, concerning to the univariate contrasts for the Stress scale, regarding the Group A between-subject factors, DASS Stress measurement 1 and DASS Stress measurement 2 were selected as within-subject factors for the comparison. In this group, it represents the training phase, and we obtained F(1,8) = 23.25; *MSE* = 9.1; *p* = 0.00. Conversely, with reference to Group B between-subject factors, DASS-Stress measurement 2 and DASS-Stress measurement 3 were selected as within-subject factors for the comparison, in the sense that this group represents the training phase, and we obtained a visible improvement, but the contrast did not reach significance (*p* = ns).

In summary, we found a significant improvement between the evaluation measurements comprising the training phase (pre–post) in group 1 and with a non-statistically significant effect, but a visible improvement in the expected direction that clearly could have reached significance with more participants involved.

As a direct result of the intervention, an increase in the tendency to pay attention in the present moment of everyday life as measured by the MAAS was also observed [64] (see Figure 4). In other words, the ANOVA results (see Table 4) showed that the increase in full attention capacity after the treatment was significant in both groups, although a slight decrease was observed during the follow-up phase for group A, as was observed in the rest of the variables. The “post hoc” overall contrast showed that differences were found between the control and treatment phases in both groups. As mentioned above, despite the promising observations that we pointed out in relation to the MAAS results, in future research, we should repeat these analyses with a larger sample because, in this case, the groups’ differences were found to be significant, which may be due to the small number of participants included in the sample.

Finally, life satisfaction measured by SWLS [70] was also evaluated, taking into account that the largest score was 25 and that, in the adaptation study [73], an average score of 19.24 was reached in the youth population. In our study, the effect of the treatment in both group A and group B produced a slight improvement in this value (See Figure 5). This improvement disappeared during the follow-up phase of group A once the intervention was completed. The results of the ANOVA showed (see Table 4) that there were no differences due to the treatment, although there were differences between the groups.

The qualitative data collected in the daily logs show a high level of satisfaction. All of them expressed their interest in continuing contact among the participants by generating mutual support groups.

## 4. Discussion

Childbirth implies a change and reorganization of the family context [85] and the most common sources of parental stress are related to the child’s health. All parents want to have a healthy child with normal development. In fact, any health disorders, typically related to possible lifelong consequences of disability in many psychological and physical areas, are logically a source of parental stress [86]. There is a great variety of parenting and parental stress that may be suffered by parents of children with neurodevelopmental problems. On the other hand, the amount of special care that a child with a developmental disorder demands may generate an extra source of stress in parents, and so it can affect their well-being and quality of life [87].

Parental stress can be also related to diagnosis timing. Sometimes, at birth or early age, the child does not present any symptoms and it is during the first months/years of life when the ASD symptoms emerge, and subsequently, parental stress emerges too. There is a range of emotional states that parents can present when the ASD symptoms arise. Climent Giné [88] describes the characteristics of these situations as a strong psychological and emotional impact, the process of adaptation and redefinition of family functioning, changes in the couple’s relationship and the need for help and advice. This stressful situation can be momentary and satisfactorily resolved or can remain for a long time, in which case it could cause a more serious post-traumatic stress anxiety disorder.

Most parents find themselves in unfamiliar situations with which they are not able to cope. One of the most well-known possible supports is providing them with quality, scientifically proven information about ASD [89,90]. However, the diagnosis news, the uncertainty about the future of their son or daughter and the increased demand for raising them increase the stress. Parents need additional strong support to cognitively restructure the situation, assume their new reality and develop new coping strategies. In this context, CBT with mindfulness and self-compassion-based programs has been shown to be effective in reducing stress in parents, but these types of interventions have been scarcely studied and fail to address a time window that is closer to the ASD diagnosis. The main goal of this article was to determine the feasibility of implementing brief interventions based on mindfulness and self-compassion within the framework of parental care received in community early care centers.

The results of the intervention program presented here prove a post-treatment decrease in self-reported values of anxiety, stress and depression in parents that might generate positive changes in the psychological well-being and quality of life of families, as evidenced in the literature [45,53,91,92]. It is worth noting that it is not a widespread practice in ECCs in Spain to apply therapies both to ASD children and to their parents to achieve better well-being in the family, since it is seen as a complex and unitary system. The reason that this practice is not more widely developed is due to the lack of regulation of mindfulness training for professionals and the practice itself as a health intervention. This lack of regulation has led to the emergence of multiple professionals with dubious training, which in turn generates dubious results, in some cases even considering full care techniques as pseudo-therapies. As more research is being developed that provides evidence, and the more mindfulness is understood as an integrated part of cognitive–behavioral programs such as MiCBT [23], the acceptance of this practice is gaining more space in the field of ASD intervention (for a review, see [5]).

In the present study, a short intervention program for parents of children with ASD based on mindfulness and self-compassion and with cognitive–behavioral and emotional regulation components was shown to be effective not only in quantitative but also in qualitative aspects. In relation to the quantitative data, it should be noted that this study presents a set of relevant data that, to our knowledge, have not been reported in previous studies. Regarding the mindfulness training used, background studies on parents of children with ASD who have implemented intervention programs focusing on contemplative practices have focused on MBSR with variants (e.g., [50]) or on self-compassion (e.g., [52]) or emotional regulation (e.g., [44]). In our study, we conducted mindfulness training that included both aspects of MBSR and SC, as well as emotional self-regulation and psychoeducation as a tool for social stigma reduction. In other words, we developed a more holistic and a systematic approach, taking into account previous studies and their successful interventions, all of them in a single intervention. On the other hand, concerning the data obtained, previous studies have reported the impact of contemplative training on measures of stress (e.g., [49,50,52]) and depression (e.g., [53]), but did not evaluate self-reported measures of stress, anxiety and depression in the same study. We believe that all three indicators can be informative in assessing the impact of holistic MBSR and SC-based programs conducted in the time window near the communication of the ASD diagnosis to parents.

The dimensions typically associated with the life satisfaction construct are feelings of happiness and loneliness (affective dimensions) and personal satisfaction (as a cognitive dimension). It seems that this dimension is affected by many other external variables that produce covariant effects. However, with regard to life satisfaction, it is important to note that during the treatment phases, slight positive changes are observed in both groups. However, it is also worth mentioning that the intervention is brief and is normally assessed in the context of a stressful situation. During the training, a certain psychological well-being improvement can be achieved, but it later deteriorates when contact with the group is lost and participants are exposed to new stressful situations. Therefore, it is possible that, if therapeutic care is prolonged by generating a group of supportive parents in a complementary way, the changes in their psychological well-being can only be observed by prolonging the sessions over time. We believe that by extending the periods of intervention and follow-up with psychological support, more significant changes can be produced.

The first main limitation of the present study is related mostly to the analysis of a reduced sample of participants. In future studies, we believe that it would be convenient to analyze a larger sample. Second, it would be desirable to include other measures, such as MPQ (Mindfulness in Parenting Questionnaire; [93]) or the PSI-4 (Parental Stress Inventory Fourth Edition; [94]), which we could not include in this study because both instruments were in the process of validation for Spanish samples when we performed the present research.

The relevance of the present clinical trial relies mainly on the relationships shown between mindfulness and self-compassion interventions and anxiety, depression and stress self-report indexes. The potential extent of these findings depends on the application of the mindfulness-based programs in community early care centers in Spain, mainly in a time window close to the communication of ASD diagnosis to parents.

## Figures and Tables

**Figure 1 children-08-00316-f001:**
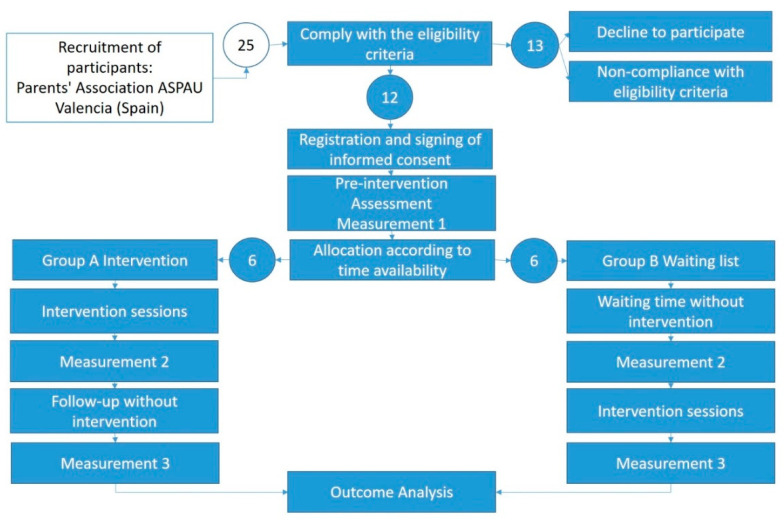
Distribution of the groups and description of the procedure.

**Figure 2 children-08-00316-f002:**
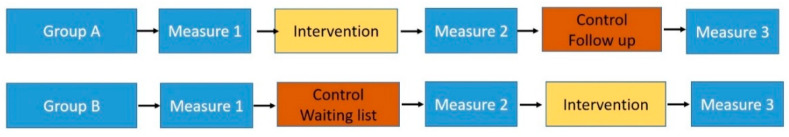
Diagram of the design of repeated measures with alternate treatment control groups.

**Figure 3 children-08-00316-f003:**
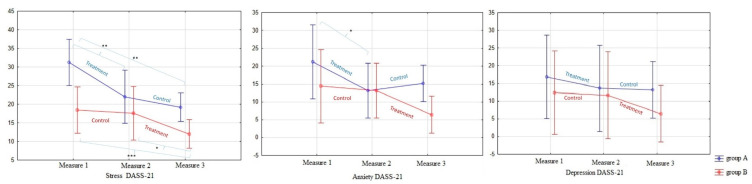
Graphical representation of DASS-21 (Stress, anxiety and depression) residual means with indication of significance of post-hoc differences. (*) *p* < 0.05; (**) *p* < 0.01; (***) *p* < 0.001.

**Figure 4 children-08-00316-f004:**
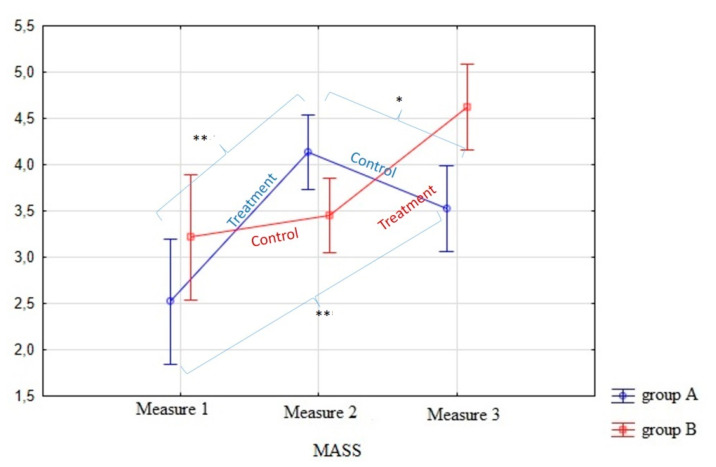
Graphical representation of MASS test results (*) *p* < 0.05; (**) *p* < 0.01.

**Figure 5 children-08-00316-f005:**
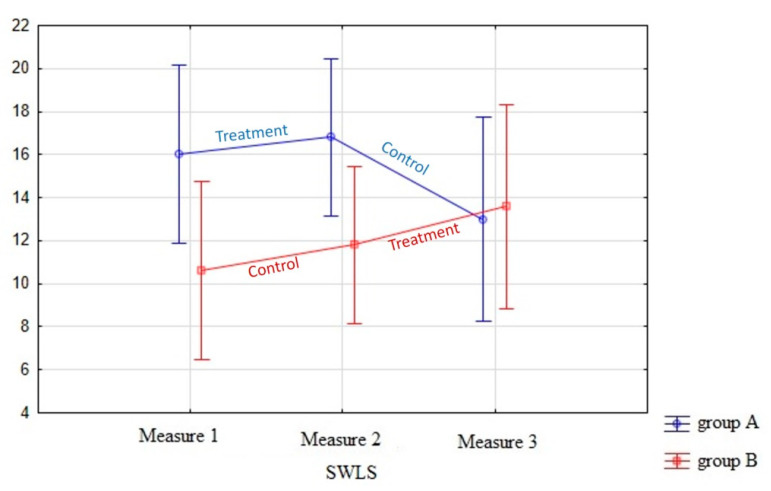
Graphical representation of SWLS test results.

**Table 1 children-08-00316-t001:** Age of the family members (parents and child with ASD) in the initial interview.

		Fathers		Mothers		Children		Level Severity		
	*N*	Age	Mean (SD)	Age	Mean (SD)	Age	Mean (SD)	1	2	3
Group A	5	40–45	43.33 (1.75)	37–43	39.83 (2.48)	3–12	5.33 (0.63)	4	1	0
Group B	5	37–48	42.25 (4.14)	33–46	40.00 (5.10)	4–10	6.90 (3.10)	4	1	0

SD: Standard Deviation.

**Table 2 children-08-00316-t002:** Topics and practices done in both mindfulness and parts of the face-to-face sessions.

WEEK *N*	Part 1: Mindfulness Session	Part 2: Psychoeducational
1	****Topic:**** Introduction to Mindfulness **Practice:** “Grape meditation”	What is autism?
2	**Topic:** Basic awareness **Practice:** Introduction to body mapping technique	Causes of autism
3	**Topic:** The importance of mindfulness to “change” our brain. **Practice:** Sitting Meditation	Intervention Methods in ASD
4	**Topic:** Self-control of thoughts. **Practice:** Introduction to yoga as a technique of mind-body integration.	Intensity or quality of the intervention in ASD. How and who?
5	**Topic:** How to “own” our own stress. **Practice:** Intermediation of automatic reactions through mindfulness	Primary and secondary school for ASD patients and parents
6	**Topic:** How to be aware of our difficult emotions or sensations before they generate consequences? **Practice:** Meditation to Calm, Allow and Accept	ASD Adolescence
7	**Topic:** Mindfulness and improving our communication **Practice:** meditation on conscious and present communication	Access to Work for ASD
8	**Topic:** Mindfulness and Self-Compassion Empathy ****Practice:**** Meditation on empathy and compassion to reduce the consequences of self-centeredness in stress	Autonomy Development and Self-Determination in ASD

**Table 3 children-08-00316-t003:** Means and standard deviations of the three scales of the DASS-21, MASS and SWLS for the two groups in the three assessed measurement times.

		Assessment Time		
		Measure 1	Measure 2	Measure 3
		Mean (SD)	Mean (SD)	Mean (SD)
DASS-Stress	Group A	31.2 (4.1)	22.0 (4.0)	19.2 (4.1)
	Group B	18.4 (7.4)	17.6 (9.0)	12.0 (3.2)
DASS-Anxiety	Group A	21.2 (6.4)	13.0 (6.4)	15.2 (4.8)
	Group B	14.4 (12.6)	13.2 (8.3)	6.4 (5.2)
DASS-Depression	Group A	16.8 (6.9)	13.6 (9.7)	13.2 (9.7)
	Group B	12.4 (14.6)	11.6 (13.7)	6.4 (5.2)
MASS	Group A	2.5 (0.86)	4.1 (0.43)	3.5 (0.61)
	Group B	3.2 (0.35)	3.5 (0.35)	4.6 (0.19)
SWLS	Group A	16.0 (3.7)	16.8 (3.1)	13.0 (3.9)
	Group B	10.6 (4.3)	11.8 (3.9)	13.6 (5.2)

MASS (Mindful Attention Awareness Scale [64]); DASS-21 (Depression, Anxiety and Stress Scale; [66]); SWLS (Satisfaction with Life Scale; [70]).

**Table 4 children-08-00316-t004:** Results of the ANOVA of all scales.

ANOVA Reapeated Measures									
	Group			Assessment Time			Assessment Time × Group		
	F	*p*	η ^2^	F	*p*	η ^2^	F	*p*	η ^2^
DASS-Stress	7.21	0.03 *	0.47	14.5	0.00 **	0.64	3.1	0.07	0.28
DASS-Anxiety	1.85	0.21	0.19	3.6	0.05 *	0.31	1.5	0.25	0.16
DASS-Depression	0.51	0.49	0.06	2.5	0.11	0.24	0.62	0.55	0.07
MASS	1.67	0.23	0.17	50.4	0.00 **	0.86	27.6	0.00 **	0.78
SWLS	2.12	0.18	0.21	0.55	0.58	0.06	4.7	0.02	0.37

(*) *p* < 0.05, (**) *p* < 0.01.

**Table 5 children-08-00316-t005:** Post hoc test (*p*): significance of the differences between measures based on the estimated marginal means.

Measures		1	2	3
1	DASS-Stress		0.00 **	0.00 **
	DASS-Anxiety		0.04 *	0.08
	DASS-Depresion		0.19	0.11
	MASS		0.00 **	0.00 **
	SWLS		0.34	1.0
2	DASS-Stress	0.00 **		0.05 *
	DASS-Anxiety	0.04 *		0.32
	DASS-Depresion	0.19		0.23
	MASS	0.00 **		0.05 *
	SWLS	0.34		0.19
3	DASS-Stress	0.00 **	0.05 *	
	DASS-Anxiety	0.08	0.32	
	DASS-Depresion	0.11	0.23	
	MASS	0.00 **	0.05 *	
	SWLS	1.0	0.19	

(*) *p* < 0.05; (**) *p* < 0.01.

## Data Availability

The raw data supporting the conclusions of this article will be made available by the authors, without undue reservation.

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
