# Peer review of "Mindfulness-Based Stress Reduction (MBSR) and Self Compassion (SC) Training for Parents of Children with Autism Spectrum Disorders: A Pilot Trial in Community Services in Spain"

_children, 2021, doi:10.3390/children8050316_

Round 1

Reviewer 1 Report

Thank you for the opportunity to review this manuscript. This is an interesting study that attempted to examine the impact of a mindfulness-based and self-compassion program on self-reported values of anxiety, depression and stress in parents of children with ASD. I value the study in that the authors addressed a new interesting topic, however, there are some concerns, include:

  1. Introduction: well-presented, could be enhanced by adding more recent studies concerning anxiety, depressions, and stress among parents of children with ASD.
  2. Method:
  • The authors mentioned in the abstract participants were 10 parents, while in the methodology 12 parents (6 participants in each group), as per Figure 1. Distribution of the groups and description of the procedure. Were there 12 or 10 participants? Were there any participants dropped later during the program? Please amend as appropriate.
  • Also, it would be helpful, if authors could provide more information about those participants, did both mothers or father of all families have participated, or some mothers and some fathers? The descriptive data of the participants are shown in table 1, didn’t indicate how many father and mothers from each family.

  1. Results:

  •  Results could benefit from a more interesting analysis between fathers and mothers.

I hope that the author/s will find this feedback helpful. Thank you again for the opportunity to review this manuscript. 

Author Response

Dear Reviewer:
Thank you for your time and comments. In section 2.8 Adherence to treatment, it is specified that two of the participants did not complete the program. One in each group. In one case it was during follow-up and the other in group B during the intervention phase. Both cases were excluded from the analyses. However, we have modified the text to make clear the final number of participants in the abstract and in the description below. We have also introduced information on the sex of the participants. Unfortunately, the vast majority of participants were mothers and we cannot make comparisons between them..
Regards

Reviewer 2 Report

Dear authors:

Please find below my comments in response to my review of the manuscript entitled, Mindfulness-Based Stress Reduction (MBSR) and Self-Compassion (SC) training for parents of children with Autism Spectrum Disorders: A pilot trail in community services in Spain. The article addresses a topic of high importance in the field of children education, which is to examine whether a brief 8-week training program using mindfulness based intervention (MBSR) and self-compassion (SC) made an effect among Valencian parents of children with ASD. However, the reviewer had major concerns.

  1. Inclusionary criteria. What were the inclusionary/exclusionary criteria for participants to participate in the study?
  2. Statistical findings. All statistical findings should be italicized when appropriate. For example, as SDs should be italicized. Make these changes throughout the manuscript.
  3. Did the authors collect treatment fidelity data? If so, the authors should describe in the methods section how treatment fidelity data were collected.
  4. Total participants. It is unclear whether all 12 participants completed all sessions. If there were participants who did not complete all sessions, this should be noted.
  5. Did the authors calculate attendance and attrition for the study? What was the attrition rate? Notably, was the study underpowered? If so, these should be included in the limitations section.
  6. Implications for research and practice. What are the implications for research and practice moving forward? What are the next steps?
  7. Directions for future research. What are the directions for future research?
  8. People first language. It is important that the authors go back and refer to parents not as “ASD parents” but rather “parents of children with ASD.” Please use people-first language throughout the manuscript.
  9. Something to consider given that the number of interested participants (versus participants who completed the training sessions) was drastically lower, is assessing the feasibility of the training itself. Given that this is a pilot study, did the authors consider collecting data to show how participants perceived the training and/or any suggestions/changes should be made to future iterations of the training? If not, this should be considered as a limitation of the study.
  10. Participant demographics table. It is recommended that the authors include a table with participant demographics (e.g., marital status; annual household income; educational background; etc.)

With these suggested edits, I believe your manuscript will be stronger and make a robust contribution to the literature.

Author Response

Dear reviewer:
Thank you for your comments. We will comment on them point by point:

1: Inclusion/exclusion criteria have been better explained in section 2.1 Procedure.
2: All aspects of formatting have been modified, both with regard to the presentation of the results in italics and the use of the decimal point.
3.- In point 2.8 on adherence to treatment, it is explained that daily activity records were used as a monitoring system. These records were delivered at each weekly session. A paragraph has been included specifying that with their analysis, not only the level of follow-up or adherence to treatment but also treatment fidelity can be determined. 
4. The summary refers to the 10 cases that completed the treatment in its entirety. As explained in point 2.8 above, one of the participants in each group did not complete all the phases (one during the follow-up phase of group A and the other during the treatment phase of group B). The summary and the corresponding section have been modified in order to achieve maximum clarity.
5. In point 2.8 it is reported that only 1 case in group A and during the follow-up and 1 case in group B during the intervention phase dropped out of the study.  The rest of the participants, who took part in the 8 face-to-face sessions, recorded more than 80% of the activities on a daily basis and those who did not, recorded more than 67% of the daily records. We have improved the wording of the section to introduce more clarity in the presentation.
6. In the discussion section, we conclude that due to the results of this intervention, programs of these characteristics should be offered systematically and at the community level.
7. In the discussion section, it is proposed as a future direction to continue developing intervention programs based on Mindfulness aimed at parents of children with ASD or at risk of developing it during the early intervention stage (2 to 4 years) and close to the time of diagnosis as this is a critical moment.
8.- The term "ASD parents" has been changed to "parents of children with ASD" throughout the text.
9. A paragraph has been introduced explaining that from the qualitative information collected in the daily non-formal records, there is a high level of satisfaction with the intervention and its results and the interest in maintaining the activity and contact forming mutual support groups.
10. Given the small number of participants, and the homogeneity in socio-economic and family variables resulting from the application of the eligibility criteria, it was decided to introduce a paragraph in the description of the participants.

Thanks for all

Reviewer 3 Report

In the manuscript entitled: “Mindfulness-Based Stress Reduction (MBSR) and Self Compassion (SC) training for parents of children with Autism Spectrum Disorders: A Pilot Trial in community services in Spain”, the authors evaluate a training on autism parents considering different scales.

            The experimental design is laudable, especially for the crossover design allowing to test within-subject, controlled effects. However, I found very hard follow several Method section. In particular:

  1. The description of participants lacks of several information;
  2. Diagnostic information about children (e.g., severity of ASD) should be provided, and might also be consider as a factor of interest, or at least as covariate;
  3. Measures (I would consider replacing with Evaluation) section is lengthy and over- detailed, the focusing the reader from the salient information for the study understanding;
  4. Conversely, data analysis is insufficiently reported. Statistical design should be clearly indicated, along with the rational supporting each of conducting analysis.
  5. Results: at first glance the two groups differ in terms of measure-1 scores for some of the investigated variables. Authors should investigate this point before entering a factorial design. In this vein, authors should also report the main effect of Group for all the ANOVAs, indeed, if significant, such an effect would reveal a major difference intrinsic to the group assignment, and potentially undermining the value of the obtained findings. This seems particularly true for Stress DASS-21 in which the lowest score of Group A is higher the highest of Group B.

            As a side note, authors should indicate in figure 4 the comparisons that are significant at post-hoc. Statistical report is incomplete, and inconsistent through out the results. For each ANOVA, F and p values should be indicated for the two main effects and for their interaction. For this latter, post-hoc comparisons should be indicated along their p value. “Univariate” test is a generic definition, a planned comparison design is fine, but its use must be justified, and the precise comparisons that are tested must be clearly indicated.

A point, not commas, should be used as decimal separator.

Author Response

Dear reviewer:
Thank you for your comments. We will comment on them point by point:

  1. This was a criticism shared by other reviewers and answered. Additional information has been introduced in the descriptive table as well as in the text.
  2. As the previous point, it has been introduced in section 2.3 In future research we would like to introduce a measure such as the AIM (Kanne et al 2014 doi:10.1007/s10803-013-1862-3) to assess the level of severity at the time of intervention. The information collected at diagnosis is not stable over time.
  3. Given the heterogeneity of measures to assess the effects of treatment we are forced to detail the measurement instruments used which may lengthen the text in the measures section.
  4. Section 2.9 specifies the statistical design. We have reread the text and introduced some small modifications in order to make it easier to read.
  5. All analyses have been reviewed and the F-values of the contrasts have been included in the text. The text has been modified, extending the explanation to improve its understanding. The graphs have not been modified because it was felt that this would make them more complex to understand.

Round 2

Reviewer 2 Report

April 10, 2021

Dear authors:

Please find below my comments in response to my review of the manuscript entitled, Mindfulness-Based Stress Reduction (MBSR) and Self-Compassion (SC) training for parents of children with Autism Spectrum Disorders: A pilot trail in community services in Spain. The article addresses a topic of high importance in the field of children education, which is to examine whether a brief 8-week training program using mindfulness based intervention (MBSR) and self-compassion (SC) made an effect among Valencian parents of children with ASD. However, the reviewer had minor concerns.

  1. Inclusionary criteria. The inclusionary criteria should be written in a consistent matter. Please check the grammar.
  2. Syntax and grammar. The authors need to go back and check all syntax and grammar thoroughly. For example, in the results section (page 9, line 372), the authors state, “We did analysis of variance…However, this analysis was already conducted; thus, this section including the Methods sections should all be written in past tense. Please fix these errors throughout the manuscript for consistency.
  3. Line spacing. Please fix the line spacing and the font of the text throughout the manuscript. Please make sure this is consistent throughout the manuscript. Notably, the spacing should be fixed for all statistical data. For example, instead of having “p=.02” it should be written as “p = .02”. Please fix all statistical data throughout manuscript. Notably, all p values should be italicized.

With these suggested edits, I believe your manuscript will be stronger and make a robust contribution to the literature.

Author Response

Thank you for your comments again:

1: The inclusion criteria section has been revised and rewritten.

2: The entire document has been reviewed for possible grammatical consistency errors. The errors detected have been corrected.

3: Line spacing errors detected have been corrected.

Reviewer 3 Report

I will stick with my former numbering:

  1. The Authors added some information, however, Table 1 reports means and SD values for both fathers and mothers, while only 1 father is included in the final sample. Please clarify.
  2. The arguments of the Authors do not relieve them of reporting diagnostic information. According to DSM-V, this should be indicated in terms of the level of severity: "Proposed DSM-5 autism spectrum criteria includes three severity classifications: Level 1 (“Requiring support”), Level 2 (“Requiring substantial support”), and Level 3 (“Requiring very substantial support”) (American Psychiatric Association 2012)." This seems parallel to what the Authors refer to as grade. Even if so, such information must be reported for all children (e.g. supplementary table) or in table 1 in the text.
  3. Using many instruments implies that all of them must be described, but not over-detailed. Once again, the Measures section is lengthy, and the title should be revised.
  4. The paragraph Data Analysis is very problematic. It must contain a concise yet comprehensive description of the statistical design, which in the original version was indicated as a two-way mixed ANOVA, with TIME as within-subject factor and GROUP as between-subject factor. A post-hoc analysis must follow. Such a design should be replicated for all measures of interest.The current version of this section confounds the factors, does not explain the whole model (two main effects and one interaction), and adds unnecessary explanations. Please restructure this section, possibly with the assistance of a statistician.
  5. Following point 4, Results are problematic as well. They should faithfully mirror the data analysis section. For ANOVA results reporting, please refer to the APA guidelines. In general,  significant effects should be clearly reported, and related p-values should be lower than 0.05. 0.1 means that the effect is not significant (line 374 pg.10). Finally, researchers can deal with graphs with data points and, at best, three asterisks. Please, add them when significant comparisons are found, otherwise the figures would be merely qualitative. 

Author Response

Thank you for your comments again:

1: In the paragraph before table 1, (page 5 line 191 and following) it is explained that, only one member of each family participated in the program. That participation was voluntary and based on their family and work obligations. Table 1 contains the age of the family members (parents and child with ASD) collected in the initial interview to which both members of the couple attended to sign the informed consent form.

The text and heading of the table have been revised and rectified to avoid confusion.

2: The required information has been entered in Table 1 and the text has been rectified.

3: The problem we encounter is that we must mention the data of the measuring instruments in the original version and in the Spanish version used in our case. This issue makes this section a little longer than usual. In addition, there are a large number of tests to measure the degree of mindfulness, so we have been forced to justify the choice of the MASS against other instruments.
4: In order to respond to their comments, we reanalyzed the data and introduced the results of previous analyses. Perhaps this is the reason for the increased confusion. The design of the clinical trial requires a two-way ANOVA statistical analysis (time as an intra-subject factor and Group as a between-subject factor).  We understood that the observation made in the previous review referred to the possible interaction or main effects analysis of each of the two factors (time or group) and for that reason we included the previous analyses.
5: A post-hoc test table was included by estimating marginal means, indicating the p-value and marking all tables and graphs with asterisks in the cases in which the differences were significant.